# Qualitative study of patient experiences and care observations during agitation events in the emergency department: implications for systems-based practice

Ambrose H Wong [1], Jessica M Ray [1,2] Christopher Eixenberger,[1] Lauren J Crispino,[3] John B Parker,[4] Alana Rosenberg [5] Leah Robinson,[5] Caitlin McVaney,[1] Joanne DeSanto Iennaco,[6,7] Steven L Bernstein [8] Kimberly A Yonkers,[9] Anthony J Pavlo[6]

For numbered affiliations see end of article.

**Correspondence to**
Ambrose H Wong;
wongambrose@gmail.com

## ABSTRACT

**Objectives** Agitation, defined as excessive psychomotor activity leading to aggressive or violent behaviour, is prevalent in the emergency department (ED) due to rising behavioural-related visits. Experts recommend use of verbal de-escalation and avoidance of physical restraint to manage agitation. However, bedside applications of these recommendations may be limited by system challenges in emergency care. This qualitative study aims to use a systems-based approach, which considers the larger context and system of healthcare delivery, to identify sociotechnical, structural, and process-related factors leading to agitation events and physical restraint use in the ED.

**Design** Qualitative study using a grounded theory approach to triangulate interviews of patients who have been physically restrained with direct observations of agitation events.

**Setting** Two EDs in the Northeast USA, one at a tertiary care academic centre and the other at a community-based teaching hospital.

**Participants** We recruited 25 individuals who experienced physical restraint during an ED visit. In addition, we performed 95 observations of clinical encounters of agitation events on unique patients. Patients represented both behavioural (psychiatric, alcohol/drug use) and non-behavioural (medical, trauma) chief complaints.

**Results** Three primary themes with implications for systems-based practice of agitation events in the ED emerged: (1) pathways within health and social systems; (2) interpersonal contexts as reflections of systemic stressors on behavioural emergency care and (3) systems-based and patient-oriented strategies and solutions.

**Conclusions** Agitation events represented manifestations of patients' structural barriers to care from socioeconomic inequities and high burden of emotional and physical trauma as well as staff members' simultaneous exposure to external stressors from social and healthcare systems. Potential long-term solutions may include care approaches that recognise agitated patients' exposure to psychological trauma, improved coordination within the mental health emergency care network, and optimisation of physical environment conditions and organisational culture.

## STRENGTHS AND LIMITATIONS OF THIS STUDY

⇒ Triangulation of both patients' lived experiences and direct observations of clinical care allow for an multifaceted and comprehensive evaluation of an understudied topic.

⇒ Use of a system-based lens is novel in identifying mediators for agitated behaviour in the emergency department outside of individual motivations and clinical factors.

⇒ Our qualitative approach is particularly powerful for uncovering systems-based factors influencing agitation events that can inform hypothesis generation and future empirical testing.

⇒ Our clinical sites are in the same healthcare system, potentially limiting generalisability.

## INTRODUCTION

Agitation, defined as excessive psychomotor activity leading to aggressive or violent behaviour, is a symptom that frequently presents in the emergency department (ED).[1] It requires rapid diagnosis of potential causes and immediate intervention to minimise harm to the patient while maintaining workplace safety for staff. Management of agitation in the ED commonly employs the use of physical restraint, which is associated with significant injuries and sudden death.[2–4] Experts champion verbal de-escalation[5] and use of structured algorithms[6] to help the clinician determine when restraint is most appropriate and minimise its use.

De-escalation is a multistep process consisting of verbal engagement, establishment of a collaborative relationship, and facilitation of patients' ability to rapidly develop their own internal locus of control.[5 7] However, it requires significant investment in time and effort to create a genuine and effective dialogue that addresses the emotional

aspects of the situation and builds trust and empathy.[8] Implementing effective de-escalation and following structured algorithms for restraint use may be difficult[9] due to systems-based limitations at the bedside, especially in the unpredictable and fast-paced environment of the ED.[10] Clinical decisions are made under time pressure, using limited information, and with multiple interruptions and other unpredictable factors.[11] Increasing medical complexity and rising demand for emergency care have also created higher cognitive demands for ED clinicians,[12] further posing challenges in building rapport with patients at risk for escalating agitation. These challenges led to recent calls to study and manage agitation using a systems-based approach, which acknowledges that an individual patient or staff member is embedded within the larger context and system of healthcare.[13] This allows for considerations of clinical factors at multiple interconnected levels of care delivery both within and outside the ED as well as their interactions with each other to develop strategies that have broader and long-lasting impact on minimising harm.[14]

Analysis of agitation events and implications for workplace safety have begun to use a systems-based approach to develop programmes that examine care delivery across hospital units and hospital networks.[15–17] However, proposed interventions that aim to decrease workplace violence have focused exclusively on healthcare workers, and their potential effects on patients are unclear. Literature directly describing agitated patients' perspectives and engagement with the healthcare system associated with use of restraint is currently limited, especially in the emergency setting.[18] This study aims to use a systems-based approach to triangulate patients' experiences related to ED visits where they were physically restrained with direct observations of agitation events in the clinical environment. We use systems lens to identify the sociotechnical, structural, and process-related factors leading to agitation events and restraint use that can inform development of patient-centred interventions in the future.

## METHODS
### Study design and setting
This is a qualitative study using a grounded theory approach to examine experiences of ED agitated patients and how those experiences interface with circumstances and factors within the healthcare system during episodes of agitation. We used a combination of two distinct datasets: (1) semistructured interviews with individuals who were physically restrained during their ED visit and (2) direct field observations of agitation events in the clinical environment. As a team of clinicians, public health researchers, and systems science experts, we aimed to explore patients' experiences and their navigation of the healthcare system directly from the participants' perspectives. In addition, field observations assisted in triangulating[19] participant data and contextualised the patient experience within the provision of clinical care in the

ED. Our study sites consisted of a 944-bed tertiary care academic referral centre and a 511-bed community-based teaching hospital with average annual adult ED volumes of 99 000 and 62 000 visits, respectively. Both institutions are part of a large regional healthcare network in the Northeast USA. All ED sites within the healthcare network had established protocols regarding indications for physical restraint use and restrictions to minimise use of restraint unless there was imminent danger to self or others. Our previous work identified approximately 1300 unique adult visits per year to these two EDs that were associated with a physical restraint.[20] We followed the 21-item Standards for Reporting Qualitative Research.[21]

### Patient and public involvement
There were no funds or time allocated for direct patient and public involvement. However, we presented our qualitative findings to people with lived experience of serious mental illness and addiction, including peer support workers, during a weekly staff meeting at the Yale Program for Recovery and Community Health (PRCH) to perform member-checking of our results. These peer support workers are individuals in recovery from mental illness or substance use disorders and have received training at PRCH to become employed on community-based treatment teams at the Connecticut Mental Health Center and in the local community. Many of the peer support workers had previously been patients in the ED and were physically restrained during the course of their care. We incorporated their feedback in our data analysis process prior to derivation of final themes.

### Interview protocol and data collection
We performed semistructured interviews as part of a prior qualitative study regarding experiences of individuals who were physically restrained in the ED.[22] Eligible individuals were adult ED patients who had been restrained during their visit to either of the hospital sites. We identified these patients through chart review of our electronic health record for visits that contained an ED restraint order. Participants received US$50 in compensation for completion of an interview. We performed purposive sampling in an iterative fashion to recruit a group of participants with demographic and clinical characteristics that reflected the overall cohort of patients restrained in the ED at our study sites.[20] We also sampled for a range of time periods between date of last ED restraint episode and date of interview. The interview guide (figure 1 box 1) reflected previous literature on patients' experiences of coercion[23 24] and ED staff perceptions of restraint use,[25] with piloting and testing prior to use. One member of the research team trained in qualitative data collection (AR) conducted the interviews while a second team member (AJP or CM) made field notes during the sessions. Discussions lasted between 40 and 60 min and were audiorecorded on digital equipment,

**Box 1. Interview guide with individuals who have experienced restraint**

A) How would you describe your health these days or in the recent past? Do you have any ongoing health problems or chronic diseases?

B) Tell me a little bit about your experiences in the emergency department and why you were there.

C) Can you recall times where you were physically restrained during your visit? What do you remember about the experience?

D) How has the experience of being restrained affected you?

E) Overall, what do you think about the healthcare system and the people who work there? How can the system do better?

F) How do your experiences during visits and impressions of healthcare affect your decisions afterwards and on a daily basis?

**Box 2. Observation field notes guide for agitation events in the emergency department**

A) Describe any interpersonal conflict you witnessed. What were the circumstances leading to the conflict and how did it resolve?

B) Document any agitation episodes you witnessed, including nature of agitation, including particular elements of the encounter that are helpful to understand the reasons and motivations surrounding the agitation event.

C) How did the staff members engage and interact with the patient, including management, clinical decision-making, attempts at de-escalation, and attitudes or behaviors that affected the agitation?

D) What were the systems-based factors that impacted the agitation event (including physical space, staffing, equipment, workflow, etc.), and how can the system better support safety during these events?

**Figure 1** Semistructured prompts for one-on-one interviews (box 1) and observations of agitation events (box 2).

professionally transcribed verbatim, deidentified, and entered into qualitative data management software (Dedoose, V.8.2.27; SocioCultural Research Consultants, Manhattan Beach, California, USA). We obtained verbal informed consent from our participants at the beginning of each session.

### Observation protocol and data collection

We observed agitation events during adult patient (>18 years of age) visits in the ED and summarised the encounters in detailed prompted field notes[26] (figure 1 box 2) performed by trained research associates (RAs) as part of a descriptive study of restraint and sedative use in the ED.[27] RAs observed each event from a short distance away and wrote narrative descriptions on an electronic tablet device. They also held brief, 1–2 min interviews with staff members after each incident to gather reflections, collect relevant clinical information, and identify systems-based challenges related to the agitation event. We scheduled four RAs in 8-hour blocks via a randomisation tool

to encompass enrollment hours between 11:00 and 2:59 hours (for all 7 days, 80 hours per week). Eligibility included any clinical encounter that required a response from protective services personnel or episodes of patient agitation (as defined by a score ≥1 on the Severity Scale)[28] identified by the RAs during regular rounds through the clinical units. Before the data collection period, the RAs participated in orientation shifts as pairs to practice conducting observations in situ, with the lead author guiding them through the process and auditing their field notes after each observation. All field notes were entered into Dedoose for coanalysis with interview data. We obtained a waiver of consent for observations given that consent could not be practically carried out without the waiver due to the agitated state that patients may be in. In addition, there was potential for physical danger associated with agitation, and the observations posed no greater risk than minimal harm and did not affect patient care or usual clinical practice.

## Data analysis

We summarised sample characteristics using medians for continuous variables and frequencies for categorical variables. A five-member coding team used Dedoose for thematic analysis and organisation of the qualitative interview data. The coding team started with blinded de novo open coding of initial transcripts, and then created and refined the code book through line-by-line analysis of all transcripts. We routinely evaluated coding categories and definitions in depth to ensure that each coder had the same understanding, modifying existing codes and identifying additional codes through iterative rounds of group discussion. Analysis of interview data provided a framework of patient experience in ED restraint that served as a scaffold to interpret observation data. Three members of the research team coded the field notes using the codebook developed from the interview data, with additional codes generated as they emerged. The final code tree consisted of a total of 291 codes within twelve main categories. After coding was complete, the entire research team integrated the two datasets through an analytic process that used a lens of systems-based practice to generate major themes and subthemes.

## Data availability

Data for this study currently exist as a deidentified dataset stored on cloud-based (Dedoose) software. Additional unpublished data can be made available to share for scholarly activities. Sharing of the data will require a aata use agreement to be established between the requesting institution and Yale University.

## RESULTS

We obtained data saturation after 25 interviews with individuals who were physically restrained during an ED visit and 95 observed clinical encounters of agitation events on unique patients. Interviews occurred between July 2017 and June 2018, and observations occurred between the months of June to August 2017. Table 1 lists basic demographics and key clinical factors relevant to agitation. Most patients for both datasets were white males and chief complaints included both behavioural (psychiatric/mental health, alcohol/drug use) and non-behavioural (medical/trauma) concerns. Within the observations, approximately 66% of agitation events resulted in use of physical restraints. Qualitative analysis identified three primary themes related to systems-based implications for agitation events in the ED: (1) pathways within health and social systems; (2) interpersonal contexts as reflection of systemic and structural forces on behavioural emergency care; and (3) systems-based and patient-oriented strategies and solutions. Table 2 provides a summary of each theme as well as their subthemes, concepts and definitions. Key quotes from patient interviews and observations of agitation events illustrating corresponding subthemes and concepts are presented in table 3. We highlight major findings from each theme below.

**Table 1** Patient characteristics and attributes

| | No (%) | |
| --- | --- | --- |
| **Characteristic** | **Interview participants (n=25)** | **Observation patients (n=95)** |
| Gender | | |
| Male | 17 (68) | 59 (62) |
| Female | 8 (32) | 36 (38) |
| Race | | |
| White | 18 (72) | 54 (57) |
| Black | 7 (28) | 29 (31) |
| Other | 0 (0) | 12 (13) |
| Ethnicity | | |
| Non-Hispanic | 19 (76) | 78 (82) |
| Hispanic | 6 (24) | 17 (18) |
| Age (years) | | |
| 18–29 | 5 (20) | 18 (19) |
| 30–44 | 9 (36) | 32 (34) |
| 45–54 | 7 (28) | 18 (19) |
| ≥55 | 4 (16) | 27 (28) |
| Triage chief complaint | | |
| Alcohol/drug use | 7 (28) | 36 (38) |
| Psychiatric/mental health | 3 (12) | 22 (23) |
| Medical/trauma | 6 (24) | 18 (19) |
| Neurologic/cognitive | 2 (8) | 8 (8) |
| Multiple | 7 (28) | 11 (12) |
| Reported reason for escalation of behaviour | | |
| Alcohol use | 7 (28) | 14 (15) |
| Drug use | 3 (12) | 17 (18) |
| Mental health issue | 6 (24) | 23 (24) |
| Alcohol/drug use and mental health issue | 6 (24) | 10 (11) |
| Confrontation with personnel/staff | 3 (12) | 31 (33) |

## Pathways within health and social systems

Agitation events were symptomatic of both the events related to a patient's pathway to the ED and the larger pathways through health and social systems that patients encountered during their daily lives. Many patients described circumstances immediately prior to their arrival in the ED as the primary reason that either led to their agitation or exacerbated their agitated behaviour during their visit. They were often coerced to go to the ED for evaluation, inducing feelings of anger, frustration and fear due to loss of control and self-determination. Interactions with law enforcement or emergency medical services in the field may have exacerbated these negative feelings in patients and led to further escalation in agitation: 'Patient was waiting for psychiatric evaluation. He said he was scared, stood and became visibly distressed when he saw the state troopers who were far away and visiting a different patient. He stated that he felt like he

**Table 2** Taxonomy of concepts describing agitation events and implications for systems-based practice in the ED

| Theme | Subthemes | Concepts | Definitions |
|---|---|---|---|
| Pathways within health and social systems | Entry into ED | ▲ Coercion affecting their ED visit experience | Being forced to come to ED against their own will or volition (eg, due to overdose, not knowing how they ended up in ED, family pressure, police) |
| | | ▲ EMS and police treatment in the field | Interaction with law enforcement or prehospital services can positively or negatively impact behaviour/ perspective in ED, or prevent unnecessary ED visits if contructive engagement occurs |
| | | ▲ Disagreement in concerns for behavioural disorders or safety | Patients coming to ED for their perceived need or complaint at odds with treatment course that providers choose to focus on or provide due to behavioural or safety concerns related to pre-arrival circumstances, with negative consequences |
| | Contexts that lead to ED use/ misuse | ▲ Social networks or lack thereof | Connections with family/friend support systems are protective, while social isolation and interactions with negative influences can trigger poor choices leading to ED visits |
| | | ▲ Ineffective/suboptimal outpatient treatment | Frustration or inadequacies, especially with mental health, alcohol/drug use, or chronic pain/medical issues, leading to ED visits |
| | | ▲ Barriers to accessing resources | Challenges in connecting to social services, including shelter, clothing, and food, as well as to medical resources, including care coordination, pain management, timely follow-up with health system |
| Interpersonal contexts as reflections of systemic stressors on behavioural emergency care | Perceived staff behaviours and responses to demanding work conditions | ▲ Positive sentiments and professionalism | ED health workers seen in a positive and constructive light during interactions with individuals, especially in context of patient-centredness and provision of respect/dignity despite significant challenges and in the face of violence or threats from patients |
| | | ▲ Staff burn-out and self-prioritisation from prolonged/repeated exposure to stress and violence | Behaviours interpreted as motivations from staff self-preservation or lack of compassion/concern for their patients, often due to systems failures or challenges, sometimes leading to feelings of empathy for staff as a result |
| | | ▲ Systemic discrimination, stigma and bias | Unfair or unequal treatment and premature closure motivated by judgement of race/ethnicity, gender, or social determinants (lack of shelter, alcohol/drug use, education) due to larger structural or institutional forces |
| | | ▲ Abuse of power dynamic and misuse of power | Violation of authority over patients, to the point where there is relish in enjoyment of power to intentionally harm or punish in retaliation of violence or assault |
| | | ▲ Lack of tolerance or capacity for non-coercive means to manage agitation | Some require behavioural methods to cope with stress in the ED (eg, pacing, yelling, seclusion, de-escalation) but ability to implement those methods is low, or staff view expression of stress as a signal to use coercive measures due to competing demands |
| | | ▲ Suboptimal behavioural care in emergency care system | Staff members lacking knowledge, expertise, training, or capacity in optimally managing behavioural symptoms in the ED |
| | Discrepancy between desired treatment and perceived actual treatment | ▲ Insufficient transparency and insight | Inadequate or unclear communication/guidance due to lack of time or resources, especially when patients disagree with plan/disposition |
| | Patients' and staff's personal motivations and desires to work within a flawed system | ▲ Changes to cognition or personality when acutely ill or intoxicated | Recognition that their behaviour was different from their normal self when they were in crisis, and that it was out of their own control or ability to be rational at that time; 'I was not myself' |
| | | ▲ Learning to navigate the system | Working or negotiating within the system to achieve personal goals, both by patients to avoid negative outcomes and by health workers to minimise personal harm |
| | | ▲ 'I don't need to be here' | Patients can perceive no medical need or logical reason to be in the ED but were coerced to enter or stay by medical personnel, inducing psychological stress due to this cognitive dissonance |

Continued

**Table 2** Continued

| Theme | Subthemes | Concepts | Definitions |
|---|---|---|---|
| Systems-based and patient-oriented strategies and solutions | Patient-centredness | ▲ Prioritise behavioural techniques and minimise coercion | Requests to first attempt de-escalation and formation of therapeutic alliance through demonstration of compassion, dignity and respect |
| | | ▲ Customised and individualised treatment plan and disposition | Involve patients as much as possible throughout visit in diagnostic and therapeutic decisions (eg, restraint placement, discharge planning, decision-making capacity) to create care plans that suit unique needs of each individual |
| | Culture and system changes | ▲ Creation of links in care delivery through pathways in health system | Improve cohesion/coordination and care transitions between various components of the agitation delivery network (psychiatric units, ED, police, prehospital services, outpatient offices) to minimise negative outcomes and use of coercion |
| | | ▲ Logistical/environmental changes to facilitate ED behavioural care | Structural and physical improvements in hallway spaces, ambulance bay, policies regarding valuables/personal property, staff communication models to protect patient respect/privacy/rights |
| | | ▲ Team-based care delivery model | Create shared mental model among different members of the ED healthcare team (nurses, providers, protective services) to ensure unified and holistic approach to agitation management |
| | | ▲ Support/training for ED health workers | Specific interventions targeting frontline staff who interface with restrained patients to facilitate/encourage empathy and prevention of burn-out |

ED, emergency department.

could not trust them because he was put in handcuffs by them when he was at the courthouse earlier that day.' (Observation 31) For patients who arrived of their own accord, discrepancies between their primary concerns regarding their perceived needs and ED staff priorities on safety and addressing behavioural concerns could lead to significant conflict and tension: '[The patient] wanted to leave against medical advice. Nursing staff told him that he was not sober enough to sign himself out and that he was too intoxicated to be discharged. He then complained that he was having chest pain and that nothing was being done about it. 'I came here because I feel like I'm having a heart attack and a pneumonia and what are you doing to help me…nothing! You just care about me having a few drinks!' (Observation 23)

Both observations and interviews uncovered that agitation events were often a symptom of larger structural barriers that patients faced to access proper healthcare and social services in their daily lives. Their socioeconomic disadvantages and struggles with chronic medical or psychiatric conditions created a perpetual downward spiral of suboptimal outpatient treatment, social isolation and housing insecurity that led to frequent ED visits, frustrating themselves and staff caring for them. One participant remarked, 'My stepfather put me in the street when I was 19 because I was out drinking and smoking after my mother died. Before long, you realize, 'I'm a homeless person. This is how I live.' I would fall asleep on the park bench, cops would pick me up all the time and bring me to the hospital. Some days I couldn't move because of the emotional pain, sadness I felt about my life. I couldn't get myself together enough to even get psychiatric help.' (Interviewee 13)

### Interpersonal contexts as reflections of systemic stressors on behavioural emergency care

Interactions that became contentious or confrontational between staff and patients did not occur in isolation, but rather reflected increasing challenges facing the emergency care system as a whole. For example, a rising volume of behavioural visits exposed staff to increasing rates of verbal abuse and physical violence: 'One technician sitting with several psychiatric patients at the same time asked [the patient] to 'please watch your mouth' and was visibly upset at being insulted multiple times. She yelled back racial slurs at the technician. He did not disengage and yelled back at the patient. This further agitated the patient and the aggravated technician stormed off the scene.' (Observation 60) These systems-based challenges overwhelmed the ability for individual staff members to dedicate enough time and resources for effective de-escalation, leading to suboptimal behavioural care and undermining patient-centred approaches to managing agitation: 'Patient had attempted suicide, and was given activated charcoal, but would not settle down and remain still. The prospect of being restrained further agitated the patient. Patient said, 'Please, I'm begging you, I can't be restrained! I'm not going to hurt anyone, I swear.'

**Table 3** Key quotes demonstrating themes

| Theme | Representative quote from interviews | Representative quote from observations |
|---|---|---|
| Pathways within health and social systems: Entry into ED | I was tellin' the cop in my house, you can't tell me what to do, blah blah blah. He didn't like that. He was tryin' to get my neighbor to file charges against me, which my neighbor did not do. So he called the ambulance, and they forced me into the ambulance. They brought me to the hospital. I tried not to stay, but after I tried to leave, they put me in restraints. (Interviewee 6) | The patient was said to be in the hospital the day before for phencyclidine (PCP) use. Today his siblings brought him by and he was extremely angry about being in the emergency department. (Observation 5) |
| | The medics' aggressiveness that they use. I had my experience when they shoved this brace on my neck because I told 'em my neck was hurting when I fell. They just shove it on so rough, like punishing me. I said, 'Is that necessary?' They don't listen. I don't know. I don't know what to say. That's how exhausted I am about it. (Interviewee 18) | He yelled, 'I have a fractured wrist and it's handcuffed here!' Patient appeared frustrated that he was placed in cuffs even though he had called the ambulance himself and was getting more agitated because of this. (Observation 21) |
| Pathways within health and social systems: Contexts that lead to ED use/misuse | I lost my truck in a car accident. I had to answer for not having a license and no insurance and no registration. I was kind of living in my truck at that point, so I went from that right into the street. I slept outside last night. The shelters they have bed bugs, and there's a whole plethora of people with issues there. (Interviewee 24) | The patient made many threats toward staff about punching them in the face. He however did not show any violent behavior toward staff. The patient became very tearful at the end of the encounter and blurted out 'My father used to rape me in my ass when I was six years old' and 'No one is helping me with my pain here' pointing to his head. (Observation 76) |
| | 'In the office, it's fine like here is the secretary, and my therapist, and my doctor. They're awesome, but it's like trying to get them on the phone, e-mail them, get a hold of them when I'm in a bad state and really need help right away, anything like that, it's like pulling teeth. They basically just tell you to go to the emergency room, so what else am I supposed to do?' (Interviewee 25) | 'I can't be waiting here for f*-ing hours. I need to get myself a ride home now. Otherwise I won't have one.' The patient was concerned that holding her there would lead to her being on the street since she was homeless, which was the reason she was picked up and brought to the emergency department to begin with. (Observation 47) |
| Interpersonal contexts as reflections of systemic stressors on behavioural emergency care: Perceived staff behaviours and responses to demanding work conditions | I know that the staff kind of have it hard in the emergency department, but they let that interfere with their care with patients. They lose interest and a lot of them don't really care and they lose compassion, assume I'm drug seeking or making it up. (Interviewee 10) | The resident acknowledged the patient's concerns about whatever happened prior to arrival. He assured the patient that the medic was no longer going to be a part of his care. Once the patient was left alone with just the doctor, he admitted to drinking alcohol stating 'I know what I need, I know I need to be in the drunk tank and sober up.' (Observation 64) |
| | It's on my chart that I'm an alcoholic so as soon as my name comes up they just instantly ship you off into the hallway and you stay here for hours and hours and hours and nobody comes and sees you. Nobody comes and takes care of you. Just because I'm an alcoholic doesn't mean I don't have issues. I have physical issues. I do have other problems, but I'm diagnosed as an alcoholic when I walk in the door and you get treated totally different. You don't get respect. You don't get any at all. It makes you uncomfortable to go. That's why I wait 'til I'm half dead to go see my primary care doctor and she yells at me all the time. I had a stroke and I didn't even go in. (Interviewee 23) | The patient was said by a staff member to have visited the ED for three consecutive days prior to this incident. The patient had borderline intellectual disability, suicidal ideation, and repeatedly banged her head against the railing of her bed. The nurse said, 'This is the third time she tried to get up and walk away from her bed. I'm not dealing with this again. She needs to be restrained.' Although the patient seemed at that moment to be docile, gentle, appeared to smile almost playfully at staff members, and was successfully redirected to her bed, the officers seemed prepared to physically restrain her; they seemed tired of having to re-direct her so many times in a row. (Observation 25) |

Continued

**Table 3** Continued

| Theme | Representative quote from interviews | Representative quote from observations |
|---|---|---|
| Interpersonal contexts as reflections of systemic stressors on behavioural emergency care: Discrepancy between desires treatment and perceived actual treatment | There's just five of those officers just waiting at the door all the time as ambulances are bringing people in cuz they're just looking for somebody to put down. I mean, you go in the emergency room and most of these drunk people, they're screaming and yelling. I can understand some of them being restrained. But I've been restrained for the least little bit—just a wise crack or something like that, and they don't like it, and they go so far over the top and hold you down and choke you out for the smallest thing.' [Interviewee 1] | The patient seemed very confused and couldn't hear properly. He seemed hard of hearing and looked around repeatedly, acting very lost. The patient was also elderly, distressed, physically non-threatening, seemed confused and scared, and it is possible that he may not have spoken and understood English properly. Prehospital staff and nurses seemed to think he was faking confusion to not be arrested for drunk driving, and did not answer his multiple attempts to ask questions. (Observation 87) |
| | I was really lost. I didn't know anything and then when they restrain you they ignore you. They don't talk to you. You can try to ask them questions, 'Why and I here? Why am I like this? Why am I restrained?' Everybody ignores you. I'm a big guy. I do demolition work. I'm tough. I'm trying to rip the restraints off of me because I don't want to be locked down and nobody's answering me. No one's talking to me. It's not fair. (Interviewee 22) | Patient seemed to be extremely agitated by being ignored by staff. Technician told me that patient was 'supposed to be on a flight to Paris' and the patient has psychosis or is delusional. Patient seemed to think that some members of staff were family members, saying that they were jealous. One nurse tried multiple times to explain that she was in the hospital and was waiting for the doctor to evaluate her but clearly became frustrated after a while. (Observation 4) |
| Interpersonal contexts as reflections of systemic stressors on behavioural emergency care: Personal motivations and desires to work within a flawed system | I don't have any felonies, but I have a lot of assaults. I grew up in [urban city], we were poor, and fighting was just a way of life and survival. It's really for me almost a first resort and not a last resort. When I see things escalating and I start to feel threatened, or—I feel fear, I'm a human being and, when I start to experience that, I do whatever I think is necessary to eliminate that. I don't know what else to say about that. (Interviewee 24) | The patient was suspected to suffer from bipolar disorder and was off his medication. The patient arrived in handcuffs and was soon put in four point restraints while in the ambulance bay. He needed to be forced to lay down on the bed while resisting being restrained. The patient repeatedly yelled, 'I need my meds right fucking now. Can I get my pills?' 'Am I real right now? I'm breathing, right?' 'I've been gone for so long now, I'm fucked up in the head. Now that I'm sober, I need fucking help.' The patient intermittently continued to yell psychotically. (Observation 65) |
| | I've also done things to avoid getting discharged, like cutting my arms. I have more of a history than just that one time. Sometimes it's like 'Okay, I need to go to the E.R. Let me cut myself a bit so that they send me up, so I can get to rehab'. More often than not, it's people that don't want to go to psych that get sent up. (Interviewee 9) | He repeatedly yelled, 'I know my rights!' 'I'm not drunk. I'm not high!' 'I have a job. I went to work today. I have six kids' 'Look at my record. I don't got a fucking record' 'I'm gonna sue the shit out of this place. I'm gonna own this hospital!' (Observation 88) |
| Systems-based and patient-oriented strategies and solutions: Patient-centredness | You gotta always remember is why you wanted to become a doctor, or nurse, in the first place. I think as long as you can remember that, as long as every doctor can hold onto that, then they'll think, 'I'll always be the best at what I do' and try to listen to us, pay attention to what we say and how we feel, even when it's hard. (Interviewee 11) | She was initially very loud and agitated, but the nurse, who was familiar with her, calmly explained that her alcohol level had to be less than 0.08 or otherwise she would have to stay in the emergency department until that point. That seemed to calm her down a lot and she agreed to the breathalyzer and when she blew over 0.11, she agreed to be calm for another hour or so until she was sober enough to make decisions. (Observation 23) |
| | The security guards were totally respectful, and they have their own personal safety and the wellbeing of their staff and the other patients to concern themselves with. I think they did what they had to—I don't think they used excessive force, and I remember it clearly. They basically said this arm is going here, this arm is going there. They talked me through it. A couple of nurses had my legs, but for the most part they had the part that really mattered, the security guards. They gave me every opportunity not to be restrained, I'll have to give em the benefit of the doubt on that. They really did. And as much as I've been in these emergency room situations, I've seen 'em have to deal with worse than me. I really appreciated that and I wish it was like that every time. (Interviewee 20) | [The patient] proceeded to verbally confront and threaten the officers, who were also standing a considerable distance away from him. The patient yelled at them, saying: 'You have no idea what I've been through.' 'I'm right here, come at me man. Come fight me.' 'I've seen some awful shit and watched for friends die, and you think it's a bad thing that I freak out over fireworks.' 'I'm ready to take all you guys out.' Eventually, one of officers who was a veteran was able to establish a good rapport with the patient, despite his persistent cursing and threats to the other healthcare staff whom he regarded with hostility. The officer calmed him down and got him to hand over his phone and he sat quietly in the stretcher after that. (Observation 54) |

Continued

**Table 3** Continued

| Theme | Representative quote from interviews | Representative quote from observations |
|---|---|---|
| Systems-based and patient-oriented strategies and solutions: Culture and system changes | I wish the psychiatrist and the doctors would talk to each other, work with each other better somehow. They can bring you to the back to the psych unit first. They have psychiatric nurses. There's usually a psychiatrist there. They have APRNs. I think these people are better prepared to attend to your psychiatric needs rather than a medical doctor who all he's gonna do is listen to your lungs, check your stomach, and that's it. That's what they call a physical. Oh, he's cleared 'cause his lungs sound clear. Well, they can send you straight to the psych unit and then the doctor can come and take a look at you there, so I'm not sitting in the hallways for hours and hours and getting me stir crazy. (Interviewee 2) | The patient was brought in by medics and was in front of the charge nurse desk. EMS had given report to the charge nurse and the patient began to yell about not wanting to be here. The EMS provider walked away from the patient without realizing he was yelling however another patient in the ambulance bay told the patient to 'Shut the fuck up'. After this the patient began yelling and swearing at the other patient so he was brought back out into the ambulance bay. While in the process another EMS provider began laughing at the patient's yelling, which cause the patient to become increasingly agitated. The medics did not seem to be aware that he caused the escalation, and the other staff members were commenting afterwards about it. (Observation 72) |
| | They don't listen, it's like they don't know how to listen properly. They have to learn how to pay attention to what I'm saying in the moment, so that things don't to be severe than what it needs to be. They put in their little reports of how aggressive, or this or that. I actually got my reports from the hospital, and it was all this nonsense that they type which was just crazy. They justify all of their behavior in their charts instead of learning how to work with me. (Interviewee 18) | While holding onto the patient's arms, nurse and officer spoke calmly to the patient to redirect her and walked her back to the stretcher. Nurse, technician, and officer attempted to de-escalate the patient as a team, informing her that she would be able to use the bathroom after prompt transfer to a hospital stretcher and movement into the ED. When the whole team de-escalated together, it seemed to put the patient at ease and she stopped yelling for a time. (Observation 90) |

ED, emergency department.

Multiple explanations by staff to her that her agitation was posing danger to herself worked briefly but she eventually necessitated physical restraint as the nurse was caring for a septic patient in the next bed.' (Observation 39)

Unfortunately, repeated exposure to acute stress and threats of violence may have led some staff to experience symptoms of burn-out and decreased sympathy for agitated patients. At the same time, cumulative and repetitive episodes of heightened stress similarly affected patients and caused them to behave in an adversarial and combative manner to exert control during exacerbations of their behavioural conditions. During an observation, a patient 'asked for water loudly when being restrained, after which the charge nurse denied his repeat requests because 'he did not ask nicely' and this further agitated the patient.' (Observation 13) However, patients learnt over time how to navigate the system to avoid negative outcomes during a visit, realising that the structures they were operating within were inherently flawed: 'The cops pick me up off the street because their bosses tell them to, or the government, or whoever's in charge. They won't want a homeless tipsy person messing up their pretty sidewalk, I get it. I really didn't need any medical treatment at all. They can't give me the things I really need—a roof over my head, a ride to my parole officer. The most they can do is a clean pair of shoes, and I'll take those. I'd get really angry and lash out at them at first, but now I've learned how to talk to doctors and I'm pretty decent at telling them what they want to hear and they let me go on my way.' (Interviewee 9)

## Systems-based and patient-oriented strategies and solutions

Both interview participants and field notes from agitation observations highlighted the need for sustainable solutions that considered the larger system of care and emphasised patient-centredness. When asked about solutions, many participants urged for compassion and use of behavioural techniques to establish rapport rather than physical restraints or chemical sedation. Although attempts to form a therapeutic alliance could ultimately fail during an encounter and added extra effort and time, those positive interactions had important positive long-term consequences on participants' otherwise fractured relationships with the healthcare system and likelihood to seek care. One participant reflected on staff showing him compassion during a period of his life when he frequently visited the ED due to poor control of his mental illness, stating that 'I would like to convey my thanks to the staff in the emergency room. They always treated me like a human being. 'Would you like a sandwich? Can we get you some juice?' Even when I was acting like a jerk because the whole world was kicking me while I'm down. It's helped me get to where I am now, even if it didn't seem like it would matter at the time.' (Interviewee 24) During observations, opportunities for patients to contribute to the treatment plan and understand the motivations underlying staff's decisions also helped tailor therapy and increase chances of success in avoiding invasive measures.

Suggested solutions and potential areas for improvement identified during observations also targeted system deficiencies that affected individual patient encounters or visits. For example, within the microsystem of the ED, improved coordination among different members of the care team could serve to create a shared mental model and increase situational awareness. During an observation, 'the patient bolted out of her bed, ran out of the ambulance bay, and ran all the way to the opposite side of the street before officers were able to catch up with her. She attempted to hit head against the stretcher and pull her hair. Afterwards the nurse and doctors noted that they did not realize the patient was an elopement risk as she was calm and cooperative during the initial evaluation. Officers overheard medics mentioning that she was suspected to suffer from bipolar disorder and was off her medication but this information was not conveyed to the rest of the team.' (Observation 66) Improving care coordination between hospital and outpatient services and creating better links in care delivery through pathways within the larger healthcare system could improve the patient experience and prevent episodes of agitation from occurring altogether. One participant remarked on his confusion regarding his medication dosing and how it caused unnecessary exacerbations of his mental illness: 'They were changing my Seroquel dosing left and right in the hospital. It finally seemed to be working, but when I got out [of the hospital], they didn't explain to my own psychiatrist why they did that, and he wanted to try newer medications with less side effects. They would not work fast enough. Within two weeks I was back in the emergency room again and mad as hell.' (Interviewee 11)

## DISCUSSION

Using a combination of patients' experiences of physical restraint and direct clinical observations in the ED, we found that agitation events did not occur in isolation but were influenced by external forces from social structures and the larger healthcare system. Disparate treatment from law enforcement or prehospital services, social marginalisation and barriers to outpatient healthcare exacerbated patients' underlying behavioural conditions and led them to inevitable bouts of agitation. These experiences spilled over into how they interfaced with emergency personnel and intensified their agitated behaviour in the ED. In addition, staff members' chronically heightened stress and anxiety from repeated exposure to violence and increasing clinical workload contributed to participants' perceived loss of compassion, systemic discrimination, and attempts to assert dominance by clinicians. Patients also developed shortcuts and self-preserving methods to navigate a flawed system, learning over time that the emergency care system was not equipped to meet their long-term social and health needs. Solutions that were deemed to be most impactful addressed fundamental system gaps in behavioural emergency care. These consisted of approaches to managing agitation that

included patient input and minimised invasive measures, creation of linkages in care delivery between hospital and outpatient care, and improved support services and coordination for team members in the ED. Identification of these systems-based stressors and effects may help policymakers, administrators and researchers to intervene on upstream targets and decrease the likelihood of agitation symptoms during an ED visit.

Our results indicated that agitated individuals were highly exposed to structural vulnerability, defined as a state of elevated risk for negative health outcomes through their interface with socioeconomic, political, cultural/normative hierarchies and societal structures.[29] This appeared to be mediated by patients' interactions with law enforcement, prehospital care and other social systems that coerced them to visit the ED. Previous work identified that agitated individuals often carried conditions that were considered stigmatising, with the majority having diagnoses of serious mental illnesses and/or substance use disorders.[20 30] In addition, vulnerable populations were over-represented in patients that presented to the emergency setting with agitation, including >30% from underrepresented racial/ethnic groups, 10% with housing insecurity and 72% from low socioeconomic status.[20 27] Unfortunately, the emergency care system reinforced these individuals' exposure to structurally mediated harm. Patients who experienced anger and resentment on top of exacerbation of their illnesses may often be held against their will, chemically sedated or physically restrained during their visit.

In this study, participants suggested potential strategies to break the cycles of structural vulnerability focused on patient-centred approaches to managing agitation. Patients could be included in the decision-making process regarding potential medication therapy. In addition, the focus of treatment could be shifted away from invasive measures and more toward minimisation of coercion or further harm. One strategy to apply these approaches that is gaining attention is the use of trauma-informed care,[31] a set of principles designed as a framework for caring for patients who have experienced harmful physical, psychological or emotional injuries (ie, trauma). Its goal is to recognise the presence of trauma symptoms, common with agitated patients, and promote a culture of safety, empowerment and healing for individuals who may have experienced trauma. For example, clinicians would recognise signs and symptoms of acute stress disorder (eg, hyperarousal, anxiety and aggression) as reactions to trauma rather than intentional or malicious intent.[32] Experts have advocated the use of trauma-informed care for violently injured persons in the ED[33] as well as those experiencing behavioural or mental health crises.[34] To better adopt trauma-informed practices, staff members who regularly care for potentially agitated patients may need additional training for structural competency,[35] an ability for health professionals to recognise and respond with self-reflexive humility and community engagement to the ways negative health outcomes and lifestyle

practices are shaped by larger socio-economic, cultural, political and economic forces within society at-large.

However, we also found that ED staff faced increasing system-based challenges that undermined attempts to promote patient-centredness during agitation events. Both patient interviews and clinical observations highlighted instances of compassion, skilled de-escalation and respect for patient dignity by staff members despite facing heavy clinical workload and repeated threats of violence and verbal insults. On the other hand, there were also instances of self-prioritisation, systematic bias, and punitive treatment, especially during times of extreme stress from demanding work conditions, overcrowding and high cognitive demand. Emergency clinicians have previously described facing a care paradox[36] and moral distress[37] when managing agitation, as their attempts to de-escalate and avoid more invasive measures may potentially threaten the safety of their colleagues and delay clinical workflow for other patients in their care. In addition, ED staff have reported symptoms of burn-out due to the frequency of exposure to agitation events and workplace violence while on shift, leading to loss of compassion and emotional exhaustion.[38–40] These symptoms may be further exacerbated by feelings of frustration and disempowerment to make large-scale changes to improve safety when agitation occurs, leading to reinforcement of negative attitudes and bias against structurally marginalised individuals.[36 41]

Recent work on mitigating bias during the management of agitation have begun to adopt a systems-based approach to derive potential solutions. This approach has allowed for a shift away from presumed fault or blame on individuals, and instead placed a larger emphasis on mediators of structural bias through institutional practices, healthcare system policies, and societal culture.[42] For example, evidence suggests that decision making based on heuristics and biases versus a more rational, methodical approach is more likely to occur under stressful conditions. A recent study found that prowhite/antiblack bias increased preshift to postshift among emergency resident physicians when the ED was more overcrowded and during times of higher patient load.[43] Strategies to minimise crowding of behavioural patients and improve staffing ratios could expand the bandwidth and opportunity for staff members to engage with agitated patients and successfully establish a therapeutic alliance.[44] In addition, creation of team-based approaches to managing agitation and inclusion of allied health professionals (eg, chaplains, social workers, counsellors) would capitalise on unique strengths from different disciplines and offload frustrations from individual staff members to a team of clinicians caring for the behavioural patient together.[45] Finally, improved behavioural, substance use, and psychiatric care transitions and organisational linkages could significantly improve the overall patient experience and prevent episodes of agitation from occurring altogether.[46 47] This may include implementation of community crisis response teams[48 49] that partner mental health professionals with law enforcement and emergency medical services to respond to behavioural crises. These efforts have successfully reduced potentially hostile and harmful forms of structural violence[50] while providing referrals to critical social and psychiatric services for marginalised communities.[51]

## Limitations

Our study has several important limitations. Patient interview data may be influenced by exclusion of individuals who declined to participate (approximately 5% of those contacted) for a range of reasons, including potential emotional distress in discussing their negative experiences during the visit and ongoing struggles with their physical or mental health. There may be potential recall bias in description of experiences and agitation events. Interviewees were more predominantly White and had lower rates of alcohol/drug use or psychiatric/mental illness chief complaints compared with patients in the observation cohort, which may influence results. Observation data may be affected by selection bias and subject to misinterpretation by the RAs due to reliance on observers enrolling agitation events and being aware of the outcomes of interest during the data collection period. Although all four RAs were trained in an identical fashion, there may be differences in interpretation of clinical events between individual RAs. However, our RAs were trained using recorded videos and subsequently coached in pairs during live observations with audits of their data collection and assurance of good inter-rater reliability (Cohen's $\kappa > 0.80$)[52] to minimise these potential limitations. Patient populations and clinical management of agitation vary between institutions, municipalities and states. Thus, our results may be subject to care delivery processes and protocols unique to our ED or local geographical region.

## CONCLUSIONS

In this qualitative study, we used a systems-based approach to triangulate patients' lived experiences of being physically restrained with direct observations of clinical encounters to uncover mediators of agitation events in the ED. We found that agitation episodes represented manifestations of structural inequity, disparate treatment and external stressors from social and healthcare systems. Our agitated patients endured forced visits to the ED against their will and hostile interactions with prehospital personnel or law enforcement prior to arrival. They also faced challenges in accessing social services and outpatient behavioural care due to structural barriers from socioeconomic disparities and cultural or societal hierarchies. At the same time, staff members attempted to provide care within increasingly demanding work conditions due to rising behavioural visits and strain on the emergency care system. This has led to limitations in their capacity to effectively de-escalate and exposed them to an increasing frequency of workplace violence, exacerbating

burn-out and reinforcing stigma and bias against marginalised populations. Potential long-term solutions may need to similarly consider and address agitation in a systemic manner to be successful. This includes application of patient-centred approaches that recognise agitated patients' exposure to harmful physical, psychological or emotional trauma and creation of support services for staff members to practice in a trauma-informed manner. Large scale changes to institutional practices and health system policies may also need to occur to improve links in the mental health emergency care network, improve access to adequate outpatient services, and optimise physical environment and organisational conditions to promote therapeutic alliance and minimise use of physical restraints. Although these efforts will take significant effort, systems-based interventions will likely have the highest impact in improving the quality of life for some of the most vulnerable and disadvantaged individuals within the emergency care system.

**Author affiliations**
¹Department of Emergency Medicine, Yale School of Medicine, New Haven, Connecticut, USA
²Department of Health Outcomes & Biomedical Informatics, University of Florida College of Medicine, Gainesville, Florida, USA
³Department of Emergency Medicine, Virginia Tech Carilion Clinic, Roanoke, Virginia, USA
⁴Department of Emergency Medicine, Coliseum Health System, Macon, Georgia, USA
⁵Yale School of Public Health, New Haven, Connecticut, USA
⁶Department of Psychiatry, Yale School of Medicine, New Haven, Connecticut, USA
⁷Yale School of Nursing, Orange, Connecticut, USA
⁸Department of Emergency Medicine, Dartmouth-Hitchcock Health System, Lebanon, New Hampshire, USA
⁹Department of Psychiatry, University of Massachusetts Medical School, Worchester, Massachusetts, USA

**Contributors** AHW, AR and JMR conceptualised and designed the overall study. AR, LC, CM and JBP performed primary data collection. JDI, SLB and KAY advised on methodological and analytic strategy. AHW, AR, CE, CM and AJP performed data analysis. All authors contributed to interpretation of the data, substantially edited the manuscript and approved of the final version. AHW takes final responsibility for the manuscript as a whole.

**Funding** This study is funded by the Robert E. Leet and Clara Guthrie Patterson Trust Mentored Research Award, NCATS KL2TR001862 and NIH K23MH126366.

**Competing interests** JMR reported receiving grants from NIH R01MD014853, Agency for Healthcare Research and Quality (AHRQ) R01HS028340, and American Medical Association Foundation outside the conduct of the study. All other authors reported no competing financial interests.

**Patient and public involvement** Patients and/or the public were not involved in the design, or conduct, or reporting, or dissemination plans of this research.

**Patient consent for publication** Not applicable.

**Ethics approval** This study involves human participants and was approved by the Yale University Human Investigation Committee institutional review board approved this study protocol (HIC# 2000020457). Participants gave informed consent to participate in the study before taking part.

**Provenance and peer review** Not commissioned; externally peer reviewed.

**Data availability statement** Data are available on reasonable request. Data for this study currently exist as a deidentified dataset stored on cloud-based (Dedoose) software. Additional unpublished data can be made available to share for scholarly activities. Sharing of the data will require a data use agreement to be established between the requesting institution and Yale University.

**ORCID iDs**
Ambrose H Wong http://orcid.org/0000-0001-7471-1647
Jessica M Ray http://orcid.org/0000-0003-3410-1507
Alana Rosenberg http://orcid.org/0000-0003-4695-5745
Steven L Bernstein http://orcid.org/0000-0002-9964-7036

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
