## [Reviewer comments · BMJ Open]

ARTICLE DETAILS

TITLE (PROVISIONAL)	A Qualitative Study of Patient Experiences and Care Observations During Agitation Events in the Emergency Department: Implications for Systems-Based Practice
AUTHORS	Wong, Ambrose; Ray, Jessica; Eixenberger, Christopher; Crispino, Lauren; Parker, John B.; Rosenberg, Alana; Robinson, Leah; McVaney, Caitlin; Iennaco, Joanne DeSanto; Bernstein, Steven L.; Yonkers, Kimberly A.; Pavlo, Anthony J.

VERSION 1 – REVIEW

REVIEWER	Im, Dana Brigham and Women's Hospital
REVIEW RETURNED	13-Jan-2022

GENERAL COMMENTS	- This is an innovative study linking patient experiences and observational data to describe systems-based challenges to managing agitated patients in the ED. This unique study proposes a framework for creating systems based solutions.- Line 378: "Previous work identified that agitated 379 individuals often carried conditions that were considered stigmatizing, with 65% having diagnoses of serious mental illnesses and/or substance use disorders" linked with reference #26, which is inaccurate.- Line 198: It is suggested that the RAs identified episodes of agitation during their rounds in the ED. Would recommend adding how agitation was defined when identifying these episodes.- What % of the patient observations (out of 95) resulted in restraints?- Consider changing subtheme "staff and system changes" to "culture and system changes."- Clarify how the concepts under the subtheme "staff and system changes" were derived during the analysis. It's unclear whether these suggested solutions came directly from the patients and the observers or if they were derived from the problems described by the patients and the observers.- Clarify whether the same patient interview dataset used for the first author's 2020 study (reference #26) was used for the current study with a systems-based approach. If yes, it would be important to note that the interviews were performed as part of a previously published study, similar to how the authors noted that the RAs were part of a descriptive study of restraint and sedative use in the ED. Did the team create a new codebook for the purpose of the current study? If
---

	so, this needs to be clarified.
REVIEWER	Beysard, Nicolas Centre Hospitalier Universitaire Vaudois, Emergency
REVIEW RETURNED	24-Jan-2022

GENERAL COMMENTS	Thank you for giving me the opportunity to review this manuscript. This is a very interesting question and the authors have used an adequate method to answer it. Find some comments below: INTRODUCTION 1) In the introduction, line 103, reference 2 is inappropriate. It refers to deaths outside the ED context. A more appropriate reference should be used. METHODS 2) The methods lacks a description of hospital protocols or guidelines (if they exist) for the management of agitated patients in the hospitals where the study was conducted. 3) Demographic data on caregivers would have been an added value: what is the proportion between new and experienced caregivers? Have they already taken training for the management of elderly patients? 4) The COREQ checklist should be completed and available. 5) For the qualitative analysis, adding a description of the coding tree should be considered RESULTS 6) Table 1 : It seems that whites patients are overrepresented among the interviewed participants in comparison with the observed patients. Similarly, among the interviewed participants, there seemed to be less "alcohol/drug use" and "psychiatric/mental health" as triage chief complaint compared to the observed patients. These elements are little/not discussed and should be considered in the interpretation of the results. LIMITATIONS 7) Line 451 : Can the authors describe the reasons why some people refused to participate? 8) Line 458 : the authors stat a good interrater reliability between RAs. Has a statistical analysis been performed (e.g. Cohen's Kappa)?
--

VERSION 1 – AUTHOR RESPONSE

Reviewer: 1
Dana Im, Brigham and Women's Hospital
Comments to the Author:

- This is an innovative study linking patient experiences and observational data to describe systems-based challenges to managing agitated patients in the ED. This unique study proposes a framework for creating systems based solutions.

- Line 378: "Previous work identified that agitated 379 individuals often carried conditions that were considered stigmatizing, with 65% having diagnoses of serious mental illnesses and/or substance use disorders" linked with reference #26, which is inaccurate.

Updated appropriate references as recommended.

- Line 198: It is suggested that the RAs identified episodes of agitation during their rounds in the ED. Would recommend adding how agitation was defined when identifying these episodes.

Added the definition – this was based on score ≥ 1 on the Severity Scale by Kowalenko et al.

- What % of the patient observations (out of 95) resulted in restraints?

Approximately 66% of observations resulted in restraints. We added this to the results section.

- Consider changing subtheme "staff and system changes" to "culture and system changes."

Agreed and changed.

- Clarify how the concepts under the subtheme "staff and system changes" were derived during the analysis. It's unclear whether these suggested solutions came directly from the patients and the observers or if they were derived from the problems described by the patients and the observers.

As explained in the methods section, all themes and concepts derived from the combined datasets of interviews and observations. Solutions were included only if both interviewees and observation field notes explicitly address them directly. We clarified this with an introductory sentence within the subsection of the results.

- Clarify whether the same patient interview dataset used for the first author's 2020 study (reference #26) was used for the current study with a systems-based approach. If yes, it would be important to note that the interviews were performed as part of a previously published study, similar to how the authors noted that the RAs were part of a descriptive study of restraint and sedative use in the ED. Did the team create a new codebook for the purpose of the current study? If so, this needs to be clarified.

Yes, we created a new codebook with de novo coding for the merged dataset. We added direct mention of the original qualitative study and clarified this in the methods.

Reviewer: 2

Dr. Nicolas Beysard, Centre Hospitalier Universitaire Vaudois

Comments to the Author:

Thank you for giving me the opportunity to review this manuscript. This is a very interesting question and the authors have used an adequate method to answer it.

Find some comments below:

INTRODUCTION

1) In the introduction, line 103, reference 2 is inappropriate. It refers to deaths outside the ED context. A more appropriate reference should be used.

Amended references to include ED contexts.

METHODS

2) The methods lacks a description of hospital protocols or guidelines (if they exist) for the management of agitated patients in the hospitals where the study was conducted.

We added a brief description regarding existing standardized protocols for restraint use within the health care system.

3) Demographic data on caregivers would have been an added value: what is the proportion between new and experienced caregivers? Have they already taken training for the management of elderly patients?

We are not sure this is applicable for our study, since this is referencing elderly patients rather than agitated patients? Please clarify and we are happy to respond as requested.

4) The COREQ checklist should be completed and available.

We completed the SRQR checklist as requested by the editor. We amended language to reflect this change.

5) For the qualitative analysis, adding a description of the coding tree should be considered

We added a brief statement regarding number of codes and buckets in our code book.

RESULTS

6) Table 1 : It seems that whites patients are overrepresented among the interviewed participants in comparison with the observed patients. Similarly, among the interviewed participants, there seemed to be less "alcohol/drug use" and "psychiatric/mental health" as triage chief complaint compared to the observed patients.

These elements are little/not discussed and should be considered in the interpretation of the results.

We added a statement in the limitations to reflect this point.

LIMITATIONS

7) Line 451 : Can the authors describe the reasons why some people refused to participate?

We added some possible reasons for refusal to participate as recommended.

8) Line 458 : the authors stat a good interrater reliability between RAs. Has a statistical analysis been performed (e.g. Cohen's Kappa)?

We added a reference to our prior study that demonstrated a Cohen's kappa > 0.80 during RA training.

VERSION 2 – REVIEW

REVIEWER	Im, Dana Brigham and Women's Hospital
REVIEW RETURNED	15-Apr-2022
GENERAL COMMENTS	Great to see the suggested feedback incorporated into this updated manuscript. Well-organized, contextualizing its important findings -- creating a roadmap for systems changes.
REVIEWER	Beysard, Nicolas Centre Hospitalier Universitaire Vaudois, Emergency
REVIEW RETURNED	24-Mar-2022
GENERAL COMMENTS	Thank you for the work done. I have no further comment. In my opinion, this work deserves to be published.